

# A geometric morphometric protocol to correct postmortem body arching in fossil fishes

Carla San Román[1,2], Hugo Martín-Abad[1,2] and Jesús Marugán-Lobón[1,2]

[1] Unidad de Paleontología, Departamento de Biología, Universidad Autónoma de Madrid, Madrid, Spain
[2] Centro para la Integración en Paleobiología (CIPb-UAM), Madrid, Spain

## ABSTRACT

Postmortem body curvature introduces error in fish morphometric data. Compared to living fish, the causes of such body curvature in fossils may be due to additive taphonomic processes that have been widely studied. However, a protocol that helps to correct its effect upon morphometric data remains unexplored. Here, we test two different mathematical approaches (multivariate regression and the so-called 'unbending functions') available to tackle fish geometric morphometric data in two exceptionally preserved gonorynchiformes fossil fishes, *Rubiesichthys gregalis* and *Gordichthys conquensis*, from the Las Hoyas deposits (Early Cretaceous, Spain). Although both methods successfully correct body curvature (*i.e.*, removing misleading geometric variation), our results show that traditional approaches applied in living fishes might not be appropriate to fossil ones, because of the additional anatomical alterations. Namely, the best result for 2D fossil fishes is achieved by correcting the arching of the specimens (mathematically "unbending" them). Ultimately, the effect of body curvature on morphometric data is largely taxon independent and morphological diversity mitigates its effect, but size is an important factor to take into account (because larger individuals tend to be less curved).

## INTRODUCTION

Body arching is a source of non-biological deformation that biases comparative settings, especially those encompassing morphometrics approaches (*e.g.*, *Clarke & Friedman, 2018*), because it leads to underestimating standard measurements of fish size (*i.e.*, body length). Although it can often be easily solved by applying basic mathematical corrections (*e.g.*, *Pan et al., 2019*), the problem is strongly accentuated when applying more powerful multidimensional methods of shape analysis such as geometric morphometrics, requiring a non-trivial solution (*Fruciano et al., 2020*; *Sotola et al., 2019*).

The most common method for assessing how to unbend body curvature in living fishes involved gradually arching an individual for taking photographs at selected degrees of body arching (*Valentin et al., 2008*). The different levels of forced curvature were thereafter assessed as shape changes *via* landmark data, resulting in a single dominant vector utilized

Corresponding author
Carla San Román, carla.sanroman@uam.es

for Burnaby's projection (*Burnaby, 1966*). Logically, this process is not applicable in fossil fishes, challenging the attainment of a vector that uniquely represents changes in curvature. Furthermore, Burnaby's method for correcting curvature is based on selecting the first dimension of a PCA (PC1) as the expected unique source of curvature. In other words, the method assumes that if curvature is the major source of variation, it will accumulate only in one dimension. However, body arching in fossil fishes alters the variance-covariance matrix, spreading randomly across PCs (*i.e.*, the effect of curvature on the superimposed data becomes latent across multiple dimensions).

When using landmark data, an alternative method to avoid the effect of body arching is the "Unbending specimens" function implemented in the TpsUtil program (*Rohlf, 2008*). This function fits a curve along a selection of landmarks that represent the body arching of an individual. The curvature delimited by such landmark configuration is projected onto a straight line according to a quadratic or cubic fit. The rest of landmarks are thereafter translated proportionally until the selected line becomes horizontally straight, representing a landmark configuration of a "straightened fish".

These mathematical methods available to correct the effect of body-curvature upon geometric morphometrics (Procrustes) shape data have only been tested in living fishes (*Valentin et al., 2008*; *Haas & Orleans, 2011*), but may not work in fossil fishes. This is because many fossil fishes exhibit a wide range of body arching in varying degrees of upward or downward curvature (*Bieńkowska-Wasiluk, 2004*; *Martín-Abad & Poyato-Ariza, 2016a*; *Cardoso et al., 2020*; *Chellouche, 2016*; *Dietl & Schweigert, 2011*; *Hellawell & Orr, 2012*; *Marramà et al., 2016*; *San Román, Cambra-Moo & Martín-Abad, 2023*; *Weiler, 1929*). This variety of body arching occurs under uncertain conditions that seem to be related to *rigor mortis*, a process that occurs when fish die and the decay of soft tissue leads to the contraction of muscles and/or ligaments, shortening and bending the backbone (*Seilacher, Reif & Westphal, 1985*; *Ando et al., 1991*; *Roth, Slinde & Arildsen, 2006*; *Chellouche, Fürsich & Mäuser, 2012*; *Pan et al., 2015*; *Viohl, 2015*; *Marramà et al., 2016*; *Martín-Abad & Poyato-Ariza, 2016b*).

Here, we adapt and test the different tools available to mathematically remove the effect of body arching on geometric morphometric data in a selected sample of exceptionally well-preserved fossil teleostean species *Rubiesichthys gregalis* and *Gordichthys conquensis*. To such end, we applied the methods to correct curvature without introducing any additional geometric morphometric distortion and tested the correlation with independent data such as body size (allometry).

## MATERIALS AND METHODS

We selected 96 individuals, that were complete, fully articulated and preserved in lateral view that belong to the species *Rubiesichthys gregalis* ($N = 63$) and *Gordichthys conquensis* ($N = 31$) from the Las Hoyas fossil site (Early Cretaceous) housed at the Museo Paleontológico of Castilla-La Mancha (MUPA). These two species of gonorynchiform fishes are easily distinguished from each other by a higher body, and shorter and fewer vertebrae in *G. conquensis* compared to *R. gregalis* (Figs. 1A, 1B), among other anatomical features (*Poyato-Ariza, 1996*).

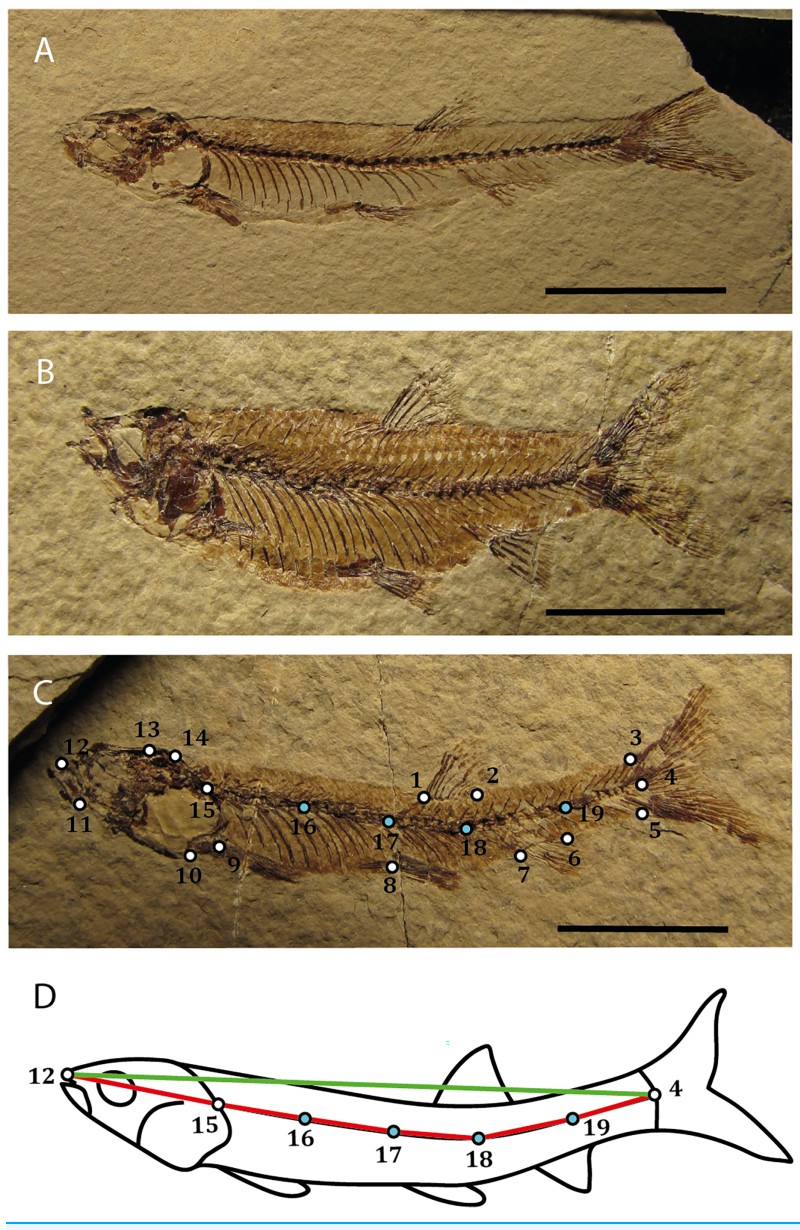

**Figure 1 Study sample and landmark configuration.** (A) *Rubiesichthys gregalis* MUPA-LH- 545a. (B) *Gordichthys conquensis* MUPA-LH-2179a. (C) Landmark configuration placed on MUPA-LH-1386a. (D) Outline drawing, landmark configuration and index of curvature (IC) measurements of MUPA-LH-1386a. Bar scale = 1 cm. Detailed landmark description are provided in Table 1 of File S3. Photo and outline drawing credit: Carla San Román.             

    The sample encompasses scale-calibrated pictures using a Canon SX510HS camera on a steady orthogonal position. A configuration of $p = 15$ landmarks were evenly placed across the whole fish skeletons using TpsDig2 (*Rohlf, 2015*), outlining the geometry of the cranium, the body and the relative position of the fins (Fig. 1C, File S3: Table 1). To avoid interobserver error, all landmarks were placed by the same person, and to reduce intraobserver error, the process of setting landmarks was repeated three times, and finally the mean landmark configuration was used.

The landmark configurations were transformed to shape data using generalized Procrustes analysis (*Bookstein, 1997*), with the packages *geomorph* and *Morpho* (*Adams & Otárola-Castillo, 2013*; *Schlager, 2017*) in the R software (v. 4.3.2, *R Core Team, 2023*).

Simply repeating the experiment developed by *Valentin et al. (2008)* of gradually arching an individual is obviously not valid for extracting a vector that uniquely constitutes curvature in fossil fishes. Interestingly, this forced experiment is naturally mimicked by the range of body arching in fossil fishes, which has been traditionally assessed utilizing an index (Index of Curvature, IC; *Martín-Abad & Poyato-Ariza, 2016b*), that is the ratio between the length from the tip of the snout to the posteriormost end of the hypurals (*i.e.*, the standard length) measured following the curvature of the vertebral column, and the same distance measured in a straight line. Given that we are performing a landmark-based study, we calculated these measurements considering the interlandmark distances resulting from the sum of the distances between landmarks 12, 15, 16, 17, 18, 19, and 4, divided by the interlandmark distance between 4 and 12 (Fig. 1D), on the Procrustes standardized data (Fig. 1D). The index varies between 0 and 1 (or alternatively 0–100%), where zero indicates no signal of curvature (body is straight) and one indicates theoretical maximum curvature.

To explore the relation between the index of curvature and the Procrustes shape data we used a multivariate regression, in order to graphically summarize the values of the index as their corresponding values of shape arching. A principal component analysis (PCA) was used to explore morphological variation in the sample, and linear regression models of each principal component (PC) against the index of curvature were performed to test the degree at which body arching could be associated (*i.e.*, randomly disseminated) across PCs (significance level = 0.05). A linear regression model of the index of curvature against the centroid size (the scalar that measures the size of the landmark configuration for each fossil—*i.e.*, the size of each fish—that is calculated as the square root of the sum of the squared distances between the landmarks; *Bookstein, 1997*) was used to test the effect of allometry and body arching. We performed Welch's test to assess statistical differences in the curvature between species (*i.e.*, between *R. gregalis* and *G. conquensis*).

The first unbending method that we applied utilizes a multivariate regression to remove non-desired measurements from Procrustes data (see *e.g.*, *Monteiro, 1999*; *Braeger et al. 2017*; *Navalón et al., 2022*). Accordingly, we applied a multivariate regression of the Procrustes data against the index of curvature (*Monteiro, 1999*), using the *procD.lm* function of *geomorph* package (*Adams & Otárola-Castillo, 2013*). The residuals of such regression are devoid of the effect of index of curvature, by definition, and their transformation back into shape data can be used in subsequent analyses (see details of R coding in Supplemental Data). We termed this first approach "Regression-unbending method".

As a second approach, we used the "unbending function" of the Tps series (*Rohlf, 2015*). This function models a curve based on a chosen set of landmarks that depict the curvature of an individual. To such end, we used landmarks 12, 4, four additionally semilandmarks (16–19) placed along the vertebral column (delimited between landmarks 4 and 15), and landmark 15 (Figs. 1C, 1D). The unbending process adjusts the positions of the remaining

landmarks in such a way that they maintain their relative proportions while aligning the selected line to be horizontally straight. To maximize the accuracy of the unbending process, the sample was subdivided in seven groups according to their IC scores, ending up in seven partitions. After the unbending process, we removed the semilandmarks that defined the curvature (semilandmarks 16–19). Finally, this new shape data was submitted again to a generalized Procrustes analysis. We denote this protocol as "Tps-unbending method".

The removed effect of body arching is visualized as shape variation across the first six PCs of the Procrustes data for the Regression-unbending and Tps-unbending methods. We calculated the Procrustes variance—squared Euclidean distances between shapes relative to the mean sample shape (*Zelditch, Swiderski & Sheets, 2012*) —in the total sample and per landmark, for the original and modified data. A pairwise contrast was devised to test differences in the total Procrustes variance of the original *versus* Regression-unbending data, and the original *versus* Tps-unbending data.

All data management and statistical analyses (except for the "unbending specimens" function performed in TpsUtil) were conducted in R software (v.4.3.2, Team, 2023). The detailed custom-code is available in the Supplemental Data.

## RESULTS

The original sample shows a mean reduction of 1.37% (IC = 0.0137) in standard length due to body arching, with the most curved specimen exhibiting a 6.43% reduction (IC = 0.0643) in length (File S1: Table 2). Despite this low length reduction, the multivariate regression of the Procrustes data to the IC shows that IC is associated with shape variance across the whole sample of the Las Hoyas' gonorynchiform fishes (Figs. 2A, 2F = 20.26, df = 93, *p*-value = 0.01, File S3: Table 3). Importantly, results of the PCA clearly show that body arching is not accumulated in a single dimension, as demonstrated by the associations of PC2 and PC5 with the IC (File S3: Table 4). The linear regression of the IC against the centroid size shows a significant correlation (F = 14.87, df = 92, *p*-value < 0.01, File S3: Table 5), the smallest specimens being the most curved (Fig. 2B). The two studied species, *Rubiesichthys gregalis* and *Gordichthys conquensis*, show similar degrees of body arching (t = 0.48, df = 57, *p*-value = 0.951, File S3: Table 6).

The PCA of the modified data for the Regression-unbending method seems to capture the presence of shape variation related to body arching in PC1, PC2 and PC4 (Fig. 3), which is also correlated with other aspects of morphology (*e.g.*, body height in PC1, relative size of the cranium and position of dorsal and pectoral fins in PC2). In contrast, the Tps-unbending method indicates absence of shape variation related to body arching in the principal PCs (Fig. 3).

The pairwise test comparing original and modified data reveals significant differences in the total Procrustes variances between original and Regression-unbending data, as well as between original and Tps-unbending data. In both cases, the modified data show lower total Procrustes variance than the original data (File S3: Tables 7–9).

Both the Regression-unbending and Tps-unbending methods's data show lower relative Procrustes variances in the extreme of the body (LM 15 and 4, Fig. 4; File S3: Table 9)
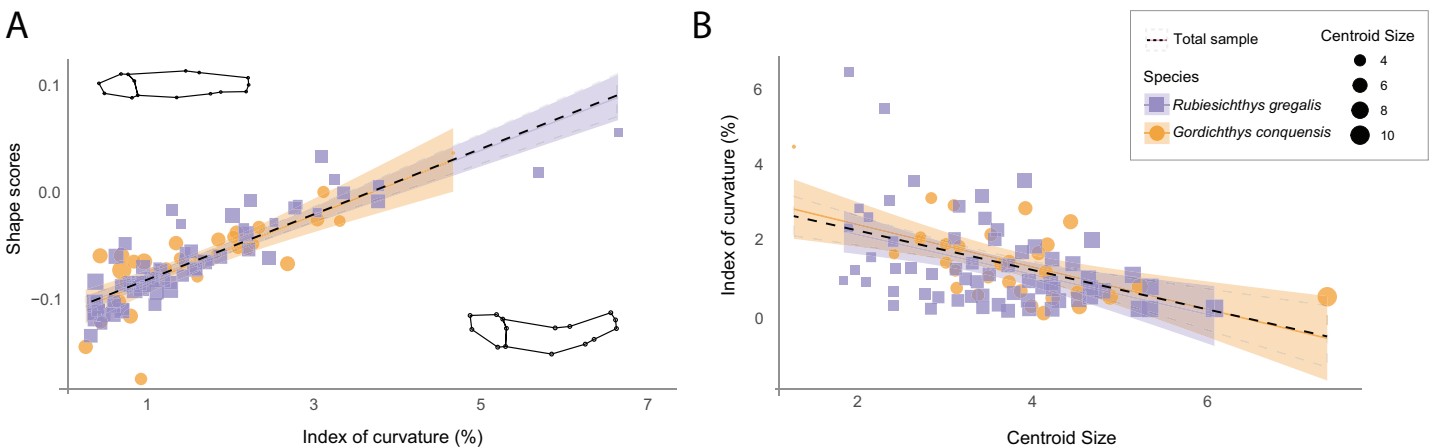

**Figure 2 Relation between shape data and index of curvature.** (A) Multivariate regression of shape scores (Procrustes data) to the index of curvature. Shape visualizations are included, where higher values correspond to curved specimens; (B) Regression of the index of curvature to centroid size. *R. gregalis* in purple squares symbols, *G. conquensis* in orange circles symbols. The size of the specimens is represented as the size of the point, following the scale of the legend. Statistical results are provided in Tables 3 and 5 of File S3.

compared to the original data. The highest Procrustes variances are located in LM 1, 2 and 8 in the original and in the modified data, accounting for body height (main difference between species).

## DISCUSSION

Post-mortem body arching by *rigor mortis* is a common process observed in fishes, that is perfectly visible in the fossil record, and it introduces important biases for statistical inference in morphometric data, particularly when this information is multidimensional (*e.g.*, landmark data). Our results on two well-known gonorynchiform fishes from Las Hoyas, *Rubiesichthys gregalis* and *Gordichthys conquensis*, show that body arching clearly affects morphological variation in different degrees, even when the curvature turned out not to be so pronounced. However, we found that body arching permeates across many dimensions of shape variation, compromising any type of multivariate method that aims at assessing biological processes based on statistical inference (*i.e.*, allometry, modularity). Thus, methods such as Burnaby's, which are expected to correct body arching assuming that it accumulates in a single (dominant) dimension (*Valentin et al., 2008*), can be ineffective in fossil fishes.

In the present study, we tested two methods for unbending fossil fishes, namely the Regression-unbending method and the Tps-unbending method. Both methods showed a clear component that dominates the variance and a significant decrease in the Procrustes variance in the modified data compared to the original. However, the Tps-unbending method outperforms the Regression-unbending method in effectively removing the curvature in fossil fishes. Namely, in the Regression-unbending method some remnants of curvature can be detected in the PCs, most likely because details of the body arching are distributed in the variance-covariance matrix and covary with allometry, thus probably comprising subsequent multidimensional analyses. The index of curvature (IC) is too simple to be able to capture the wide range of curvature in fossil fishes (*Bieńkowska-*

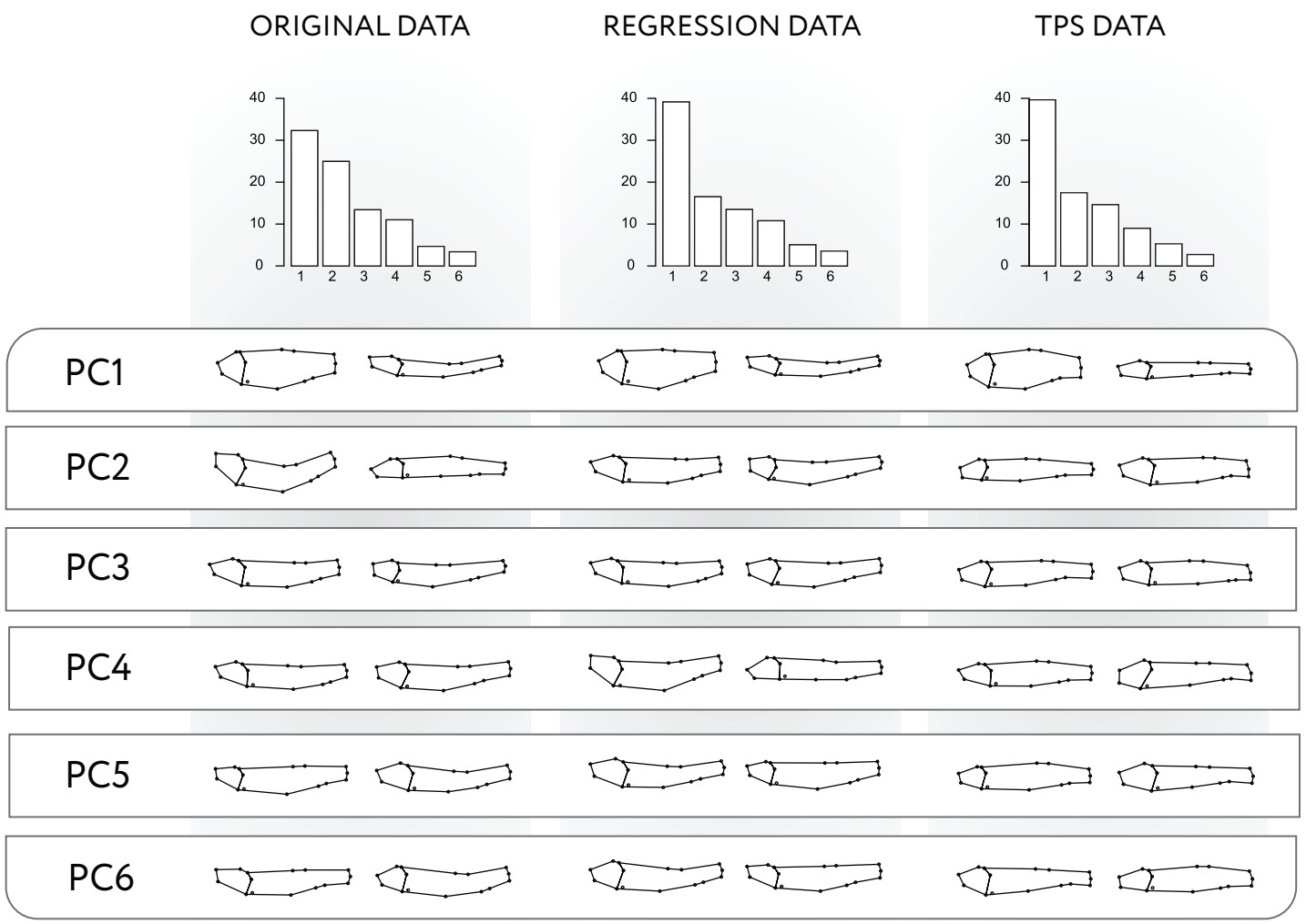

**Figure 3 PCA applied to original, Regression-unbending, and TPS-unbending data.** Procrustes variances of the first six principal components (PC) and the subsequent shape variation corresponding to each PC for the original and modified (Regression-unbending and Tps-unbending) data.

*Wasiluk, 2004*; *San Román, Cambra-Moo & Martín-Abad, 2023*). When utilizing the Tps-unbending method (*Haas & Orleans, 2011*), on the other hand, the curvature adjusts better to a cubic curve than a quadratic fit, probably because the arching of the bodies is not uniquely concave or convex, but rather it is more often irregularly sigmoidal. We believe that this way of accurately capturing body curvature explains that the variation related to changes in body arching is not observable in principal PCs. The Tps-unbending method also offers additional advantages, for instance, magnifying undetectable Procrustes variance in other landmarks, that is not attributable to geometric distortion.

To date, no study has used geometric morphometrics to explore the biological factors influencing *rigor mortis*-induced body arching in fossil fishes. Our results indicate a significant correlation between specimen size and curvature, with smaller and more slender individuals exhibiting a more pronounced curvature. This association could be explained by the degree of ossification that seems to be linked to skeletal arching (*Pan*

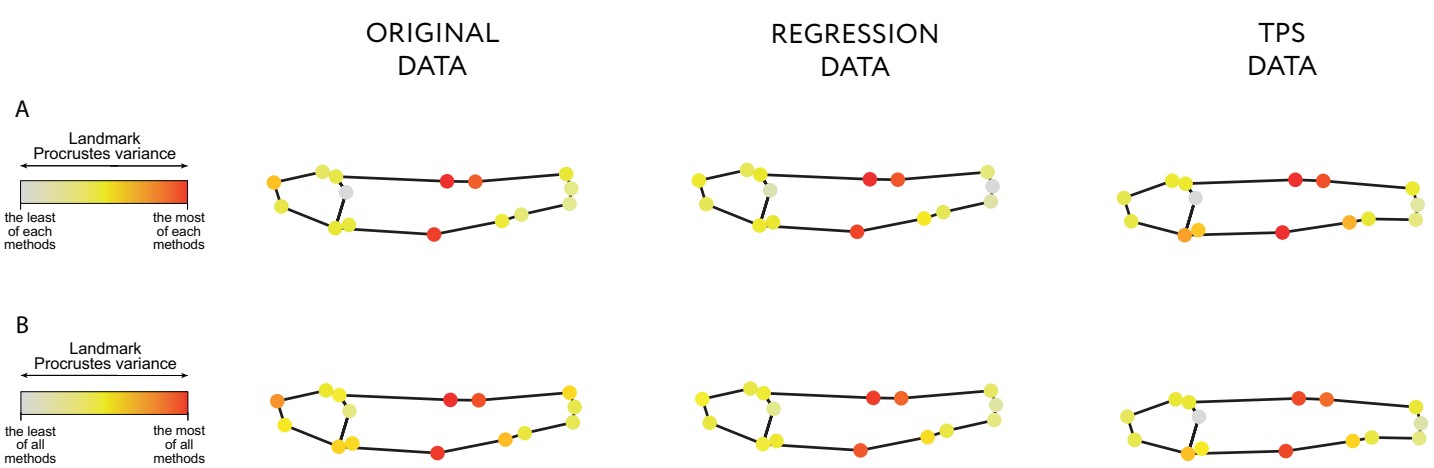

ORIGINAL
DATA

REGRESSION
DATA

TPS
DATA

A

Landmark
Procrustes variance

the least
of each
methods

the most
of each
methods

B

Landmark
Procrustes variance

the least
of all
methods

the most
of all
methods

**Figure 4 Procrustes variance per landmark.** The gradient from gray to red indicates, respectively, the minimum and maximum Procrustes variance: (A) in each data; (B) in aggregate for all samples. The mean values of Procrustes variance per landmark are provided in the Table 9 of File S3.

*et al., 2015*; *Martín-Abad & Poyato-Ariza, 2016b*). However, our analysis reveals a significant equivalence in body arching between *R. gregalis* and *G. conquensis*, suggesting that, in our specific case, taxonomic identity is not linked to the magnitude of body arching. This result counters previous results suggesting that (*Bieńkowska-Wasiluk, 2004*). It is possible that the morphological and anatomical differences between the species studied here were insufficient to significantly impact the taphonomical processes responsible for generating the curvature or even these biological differences may not be causally related to body arching, as has been seen in experimental studies with extant fishes that refuted any association (*Ando et al., 1991*). Although curvature is often attributed to taphonomic processes (*i.e.*, *rigor mortis*), there are cases where it holds biological significance. To avoid correcting curvature of natural origin (non-taphonomic), it is essential to select the appropriate landmarks for their use in the straightening protocol. For instance, in our case, the anterior point of the premaxilla and the vertebral column form an axis that marks the straightness of the organism. However, in other cases, this axis may be defined by different landmarks that must be considered after prior anatomical evaluation of the sample.

Given these considerations, our results represent a step towards correcting the effect of body arching in fossil fishes that can be extended to any particular case study. The more similar the species are (at least based in general shape morphology), the larger the effect of the curvature will have into blurring morphological and biological data, thus reverberating in lowering the resolution of morphometric studies (*Clarke & Friedman, 2018*; *Kammerer et al., 2020*; present study), particularly if statistical inference is intended. Namely, correcting this variability is advisable because it can hamper the evaluation of important evolutionary phenomena such as allometry, integration and modularity.

## CONCLUSIONS

Two different methods have been designed to correct variation associated with body arching from a landmark configuration, using two well-identified gonorhynchiform fishes from the Las Hoyas fossil site, *Rubiesichthys gregalis* and *Gordichthys conquensis*, as a case sample. A multivariate regression, based on the regression of the Procrustes data against the assessed degree of arching (Regression-unbending method) corrects such effect. However, details of arching are distributed in the matrix of variance-covariance, and thus might appear in further statistical analyses. Therefore, we recommend using the unbending function of TpsUtil (Tps-unbending method) to get rid of the largest amount of variation related to body arching. We also show that curvature largely depends on the size of the fish (smaller ones tending to be more curved), thus being an important factor to consider in order to ensure accurate and meaningful analyses. No evidence of differences in curvature were related to taxonomic identity, but this needs to be further tested with samples encompassing higher morphological and taxonomical diversity.

## ACKNOWLEDGEMENTS

The authors express gratitude to the Museo Paleontológico de Castilla-La Mancha for providing access to the Las Hoyas collection. We extend our appreciation to Dr. G. Navalón and Dr. S.M. Nebreda for their valuable discussions, which have improved the manuscript, as well as for generously sharing their R code, which was utilized to adapt certain aspects of the R code presented in this work. We want to thank the editors and anonymous reviewers for offering constructive comments which have greatly improved the quality and clarity of this manuscript.

### Funding

This work was supported by the Spanish Government under Project PID2019-105546GBI00 and the Department of Biology, Universidad Autónoma de Madrid under Project BIOUAM02-2019. Carla San Román is supported by a FPI-UAM Ph.D. scholarship from Universidad Autónoma de Madrid. The funders had no role in study design, data collection and analysis, decision to publish, or preparation of the manuscript.

### Grant Disclosures

The following grant information was disclosed by the authors:
Spanish Government: PID2019-105546GBI00.
Department of Biology, Universidad Autónoma de Madrid: BIOUAM02-2019.
Carla San Román is supported by a FPI-UAM Ph.D. scholarship from Universidad Autónoma de Madrid.

### Competing Interests

Jesús Marugán-Lobón is an Academic Editor for PeerJ.

## Author Contributions

- Carla San Román conceived and designed the experiments, performed the experiments, analyzed the data, prepared figures and/or tables, authored or reviewed drafts of the article, data Collect, and approved the final draft.
- Hugo Martín-Abad analyzed the data, authored or reviewed drafts of the article, data Collect, and approved the final draft.
- Jesús Marugán-Lobón conceived and designed the experiments, analyzed the data, authored or reviewed drafts of the article, and approved the final draft.

## Data Availability

The R code, raw data and all additional results that correspond to the tables are available in the Supplemental Files.

## Supplemental Information

Supplemental information for this article can be found online at http://dx.doi.org/10.7717/peerj.17436#supplemental-information.

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
