# Peer review of "A geometric morphometric protocol to correct postmortem body arching in fossil fishes"

_PeerJ, doi:10.7717/peerj.17436_

## Round 0.1 · original submission · Minor Revisions

I have now received three comprehensive reviews on your manuscript, all from experts in geometric morphometrics. All of them saw your paper with interest, despite mentioning its narrow focus and scope. You'll notice that all three reviewers' comments are focused on clarifying several aspects of the methods section of the paper. Several details are indeed lacking, so pay close attention to this. Again, all three reviewers mention that the interpretation of the ordination diagram is not fully correct. R2 and R3 point out that you need to provide test statistics (and ideally also an effect size measure) along with P value.

If my Castellano is correct, I think by 'retiring' you mean 'removing' or 'correcting' in L. 23.
L. 98-9: no need for a non-parametrics test here, simply use a separate-variances t test.
In Results, always provide a test statistics along with P values. P values alone don't mean much.
Remove citations of tables and figures from Discussion.

I also think this is the kind of methodological paper needed to evaluate the usefulness of alternative methods in GM. Thus, I invite authors to revise their paper in response to the reviewer's critiques.

Reviewer 1 ·

Basic reporting

The paper aims to test two different tools to remove the effect of body arching on fish fossil data using Geometric Morphometrics methods. I think the english is well written, only with some minor errors, but I am not a native speaker. I find the paper very interesting, well written and concerning an important topic. I have, however, some minor suggestions, especially in the M&M section and discussion.

First of all, I think the title cold change a little bit to make clear that the paper is about fish fossils. Reading the title, I was surprised when I first found out that it was about fossil, so I suggest changing “fishes” to “fish fossils” in the title.

Here I list some comments that I find important to improve the article:

1. Line 74. The authors mention that they digitized 15 landmarks, referring to the Fig. 1C and Supplementary table 1. By taking a look in the figure and table, however, I found out that there was four additional semilandmarks. I kept reading and them in lines 118 and 119 the authors finally described those “semilandmarks”, apparently used only for the tps unbend procedure. Is my interpretation correct? If such “semilandmarks” were used only in the tps procedure, I suggest separating them from the table 1 and maybe also from Fig 1C, to avoid such confusion. If, however, my interpretation is not correct and the four “semilandmarks” were also used in the GPA procedure, such landmarks should be cited in line 74 together with the other 15.

2. Still about the “semilandmarks”: lines 118-119 refer to them as “landmarks”, while table 1 refers to them as “semilandmarks”. By concept, semilandmarks are not fixed points, but relaxed to better fit forms. If the landmarks 16-19 were not allowed to slip to fit a curve, they are landmarks (or pseudolandmarks, following some authors) - and should be referred to as "landmarks" in table 1.

3. Line 87. After introducing the IC index, from Martín-Abad & Potayato-Ariza, 2016b, the authors stated that they “calculated the IC as the interlandmark distance resulting from the sum of distances between landmarks…”. It was not clear to me if such way of calculating was taken from Martín-Abad & Potayato-Ariza or elaborated by the authors and I couldn’t find the original article to check. Please make it clear.

4. Lines 136-137. In the first line of the results, the authors mentioned that the “sample shows a mean reduction of 1.37% in standard length due to body arching, with the most curved specimen exhibiting a 6.43% reduction in length”, citing the supplementary table 2. Reading the M&M again, I couldn’t find how did they come to these values. They explained how to calculate the IC value, that should vary from 0-1. They did not say, however, how to calculate the standard length and expected standard length. Please, make it clear in the M&M.

5. Line 138. The authors say, “Despite this low length reduction, the PCA shows that body….”, referring now to the fig. 2A and supplementary table 3. The figure 2A does not show the PCA, but the Procrustes coordinates against the index of curvature. Please correct it.

6. Line 148. The authors claim that the PCA of both Regression-unbending and Tps-unbending methods indicates absence of shape variation related to body arching in the principal PC’s, calling the figure 3. Examining the figure 3, however, I do not really agree with such conclusion. It seems to me that some PC’s in the Regression data shows some shape variation related to arching. During the discussion this conclusion appears again in line 178; in lines 184-185, yet, they say that in the Regression-unbending method some remnants of curvature can be detected in low PCs. For me, it can be detected even in the first PC’s.

7. Lines 159-160. Here the authors say that the morphological variation related to taxonomical identity is not modified in the unbent data by any of the two methods, again calling the figure 3. I can't clearly see this in figure 3, though. It would be good here a scatterplot PCA graph for the original data and both unbend methods coloring the species with different colors to see if specimens from each species group together.

8. Discussion: Overall, the discussion is good and well written. I have some suggestions regarding the conclusion about both unbend methods being equally good, and here in the discussion they actually say that the tps method is better – which I agree. So I think they should make it clear through the whole paper. Nevertheless, I think the discussion could address some more points: since the paper deals with unbend methods, that actually transform the shape that is being analyzed though a very sensitive method (Geometric Morphometrics), I figure if there is not a possibility that arching could be a natural shape variation in some cases, that is being taken off artificially. It would be nice to discuss this. Is there any risk that such methods might remove genuine shape variation? Or is there no evidence in nature of arching as a source of natural variation in fishes? Still, in line 202-203, the authors say that taxonomic identity is not linked to the magnitude of body arching. Although I agree that this is extremely important, I think other aspects besides taxonomy are also relevant. Is any morphological aspect linked or masked by body arching or unbend procedures?

Some minor comments as highlighted in the pdf attached.

Experimental design

The Materials & Methods could be a little improved, as mentioned in "Basic reporting".

Validity of the findings

no comment

Annotated reviews are not available for download in order to protect the identity of reviewers who chose to remain anonymous.

Reviewer 2 ·

Basic reporting

A useful contribution. I have only relatively minor comments.

line 16: I am not sure if one should say "random" here. The results show less curvature for larger specimens. That is a systematic effect.

line 23: "retiring" is the wrong English word. Unclear what is intended.

line 33. perhaps say "death" rather than decay?

line 44: I think "bottom" must not be the right word. Meaning is unclear.

Line 56: due to the preservation process or just a process associated with death or decay as mentioned above?

line 67: "selected" not "select".

lines 73-74: what is a "steady set"? Perhaps a tripod?

line 96: Should be "square root" - not squared root.

line 116: this sentance seems to suggest that the method was first developed by Hass & Orleans and then later put in the tpsUtil program in 2015. Hass & Orleans obviously could not comment on the funtion in the tpsUtil program unless it already existed.

line 118: "four equally spaced landmarks" - perhaps one should say semi-landmarks.

line 143 (and elsewhere): I would prefer that you also give the actual value of the statistic being tested - not just its P-value. A P-value could be quite small but unless the correlation is large the correlation probably is not of much interest.

line 199: Better to just say "To date,".

Table 9: The variance at each landmark is not usually of much interest. Can be an artifact of the superimposition method. Probably better to just give their sums for each medod.

Experimental design

ok

Validity of the findings

ok, See my comments above.

Reviewer 3 ·

Basic reporting

-Lines 53-58: could be expanded upon. The authors state “fossil fishes add complexity to body curvatures due to preservation process, but the mathematical methods available…” what are those methods? Should introduce those, especially if they are what the authors use in their analyses. I feel this is especially importance since most of the introduction focuses on rigor mortis and how/why it occurs (not totally relevant to this paper), what is relevant are the methods to correct and address rigor mortis, which is extremely relevant to this paper.

-Line 60: “Here, we test the different tools available…” again, that should be explicitly stated to set up your question more directly and to make it clear exactly what you are testing.

-I am a bit confused which species is included in which analyses (except when they explicitly state it) and whether or not they were treated as separate factors in these analyses. Just more clear text is needed on what exactly each model is testing and what was or was not included as variables (e.g., species?).

-Line 138: The authors state “…the PCA shows that body arching is correlated with shape…” by body arching, the authors mean IC? Be consistent with terms.

-Line 139 and elsewhere: appropriate test statistics (F stat, DF, T stat) need to be included along with the p-values.

-Line 141: this might be personal preference, but I typically always would use “associations” instead of “correlations” when performing regressions. There is a distinction between the two for me, correlation isn’t predictive, whereas associations is.

-Line 143: “lineal” to “linear”

Figure 2A: the X-axis is “Index of Curvature (%)” but those appear to be frequencies? Based on the numbers they have in their supplementary file.

Figure 3: In the text, the authors discuss bending only appearing on PC4 for Regression Data, however, I see some clear bending on PC1 and PC2, especially relative to the TPS data. Think it would be good for the authors to revisit that and discuss or explain.

Figure 4: The legend needs to be more detailed, I’m not entirely sure the difference between A and B. Also, the gradient as I can see it goes from white to red, not gray to red as stated in the legend.

Line 168: “even when the curvature since not to be so pronounced” should be rewritten to something like “even when the curvature turned out not to be so pronounced” unless I’m misunderstanding what they are trying to say.

Line 175-176: “and the Tps-unbending method, have proved to preform better at correcting body arching” is that from this paper or a different one? There is not citation, so I assume that’s their conclusion. If so, this is the first time it’s stated, so it should be made much clearer.

Line 200: “strong” to “significant”?

Lines 202-203: the authors state “in our specific case, taxonomic identity is not linked to the magnitude of body arching.” Which is fine, however, might magnitude influence different species differently? I.e., as stated in the introduction that geometric morphology is very complex, does a 1% IC represent the same geometric changes in different fish species?

Experimental design

-Line 67: “We select 96 complete individuals…” How? Were they randomly drawn from a larger pool of samples? Or are there only 96 samples available? Also, should be “We selected 96…”

Line 73- How was the camera held steady set? On a tripod or a build platform?

Line 74- who set the landmarks? Was it only one person? “reader bias” in landmark placement can be very influential, so at least clarifying that one person exclusively placed landmarks is important. If that wasn’t the case and multiple people did it, justification is needed for how that wouldn’t influence their findings or introduce bias.

Line 78- I believe the citation for R is incorrect and should be “R Core Team 2023” and addressed as Program R.

Lines 80-87: this info don’t seem relevant to me, especially since it seems their IC measurement is completely different? Perhaps that portion can be shortened, and they can add a clearer explanation of how to interpret their IC values because with some silly math I did and their reported numbers, it seems the 0 and 1 bounds no longer apply here ass cited on line 86?

Validity of the findings

No Comment.

---

## Round 0.2 · accepted · Accept

Thank you for carefully addressing reviewer's comments and also my own. I'm glad to recommend it for publication as is.